# Classification und Treatment Algorithm of Small Bowel Perforations Based on a Ten-Year Retrospective Analysis

**DOI:** 10.3390/jcm11195748

**Published:** 2022-09-28

**Authors:** Flurina Onken, Moritz Senne, Alfred Königsrainer, Dörte Wichmann

**Affiliations:** 1Department of General, Visceral and Transplantation Surgery, University Hospital of Tübingen, Hoppe-Seyler-Str. 3, 72076 Tübingen, Germany; 2Department of Anesthesia and Intensive Care Medicine, University Hospital of Tübingen, Hoppe-Seyler-Str.3, 72076 Tübingen, Germany

**Keywords:** small bowel perforation, iatrogenic small bowel perforation, short bowel syndrome

## Abstract

Background: Small bowel perforations are a rare diagnosis compared with esophageal, gastric, and colonic perforations. However, small bowel perforations can be fatal if left untreated. A classification of small bowel perforations or treatment recommendations do not exist to date. Methods: A retrospective, monocentric, code-related data analysis of patients with small bowel perforations was performed for the period of 2010 to 2019. Results: Over a 10-year period, 267 cases of small bowel perforation in 257 patients (50.2% male and 49.8% female; mean age of 60.28 years) were documented. Perforation’s localization was 5% duodenal, 38% jejunal, 39% ileal, and 18% undocumented. Eight etiologies were differentiated: iatrogenic (41.9%), ischemic (20.6%), malignant (18.9%), inflammatory (8.2%), diverticula-associated (4.5%), traumatic (4.5%), foreign-body-associated (1.9%), and cryptical (1.5%) perforations. Operative treatment combined with antibiotics was the most commonly used therapeutic approach (94.3%). The mortality rate was 14.23%, with highest rate for patients with ischemic perforations. Discussion: An algorithm for diagnostic and therapeutic steps was established. Furthermore, it was found that small bowel perforations are rare events with poor outcomes. Time to diagnosis and grade of underlying disease are the most essential parameters to predict perforation-associated complications.

## 1. Introduction

Intestinal leaks and perforations are defined as a loss of integrity of all wall layers with the consecutive leakage of intestinal content. The cause and etiology, as well as the epidemiology, of small bowel perforations are not well-described. Kimchi et al. described an incidence of 1 case in 350,000 people per year of non-traumatic small bowl perforation [1]. Various causes of perforation have been described, e.g., iatrogenic, traumatic, infectious, malignant, and diverticulum-associated [2,3,4,5]. We found no recent publication describing the distribution of perforations and their outcomes in a large population.

According to the location of the perforation, intestinal secretions may retro- or intraperitoneally leak and result in remaining clinical symptoms [2,6]. Covered perforations of the duodenum are characterized by diffuse pain with positive renal positional palpitation when nerves are affected [7]. Free perforations into the abdominal cavity show clinically conspicuous symptoms of an acute abdomen and peritonism. Untreated, uncovered SBP leads to multi-organ failure and patient death from sepsis [2]. The timing of diagnosis of an SBP can be decisive for prognosis [8,9,10,11]. Short bowel syndrome (SBS), an associated complication of SBP, is a serious diagnosis itself, with increased morbidity and mortality and a significant reduction in the quality of life [3].

Cross-sectional imaging through computed tomography (CT) has been established as the gold standard to diagnose small bowel perforations [12,13,14]. According to the localization, genesis, and extent of the perforation, therapeutic measures may be endoscopic, laparoscopic, or open surgical [7,15,16,17].

SBPs are a rare diagnosis compared with esophageal, gastric and colon perforations [2,12]. No classification for small bowl perforation has been published yet, so comparisons of frequency and outcome in the different publications are not possible.. The aim of our retrospective study was to define different types of small bowl perforation and their outcomes.

## 2. Materials and Methods

The local ethics committee of the University Hospital of Tübingen, Germany, approved this study (7 August 2020; AZ: 489/2020BO). The study is registered at Clinicaltrails.com (NCT05471999). All patients who received treatment for SBP between January 2010 and December 2019 were considered for the study. Data acquisition was performed based on an ICD-10 code-based query. The following codes were used: K 26., 27.1, 28.1, 55.0, 57.0, 57.4, 57.8, 63.1, and 63.3. ICD-10 code S31.83 was used in combination with S36.4 (small bowel injuries) to search for iatrogenic intraoperative injuries.

Informed consent for data analysis was prospectively obtained from all individual participants. Patients’ records were retrospectively analyzed. The inclusion criterium was an age of more than 18 years, and younger patients were excluded from the analysis. The search results are shown in Figure 1.

Analysis included parameters of the patient’s history, symptoms and clinical features, diagnostic strategies, therapeutic strategies, and their outcomes. The primary endpoint was the SBP-associated 30-day mortality, and secondary endpoints were the rate of associated short bowel syndromes, the length of hospital stay, and the general morbidity rate.

Data analysis was performed using MS Office Excel 2019 (Microsoft Corporation) and SPSS 28 (IBM, SPSS Statistics, New York, NY, USA). The following quantitative and qualitative tests were conducted: Χ^2^ test, Kruskal–Wallis test, and Mann–Whitney U test. The level of significance was 5%, and the results were statistically significant if *p* < 0.05.

## 3. Results

Data of 267 cases (male *n* = 134, female *n* = 133, and mean age of 62.6 years [19–102]) of SBP in a period of 10 years were analyzed. Eight categories of causes of SBP were found and are presented in Table 1 according to the number of events.

Iatrogenic perforations (category 1) were found to have taken place in laparotomies (*n* = 89), laparoscopies (*n* = 10), endoscopies (*n* = 8), and interventional punctures for drainage tube placement (*n* = 5). Ischemic SBP (category 2) occurred in 55 patients. The malignant diseases causing SBP (category 3) were lymphoma (*n* = 9), gynecological cancers (*n* = 9), GI cancers and tumors (*n* = 14), metastases of skin cancers (*n* = 3), metastases of kidney cancers (*n* = 4), peritoneal cancers, and metastases of lung cancers (*n* = 2). These three etiologies of SBP caused nearly 80% of all reported perforations and 91% of the reported perforation-associated deaths.

In 189 cases, previous GI surgery was documented (70.8%). The most common clinical symptoms of patients with SBP were abdominal pain (45.3%), vomiting (11.2%), fever (3.1%), and suspected fluid quality in placed drainages (8.6%). Twenty patients suffered from sepsis (5.2%), and peritonitis was found in 35 patients (9.1%). Simultaneous bleeding was found in 7 patients (1.8%). CT scan resulted in the diagnosis of SBP in 86.2% of patients. In the other patients, the suspect quality of drainage tubes was cause for the suggested perforation. The localization of small bowel perforations was documented as follows: 4.87% in the duodenum, 37.83% in the jejunum, and 39.33% in the ileum; no localization data were available in 17.98% of patients.

The therapeutic procedures for SBP are listed in Table 2. The main therapeutic procedure for patients with SBP in this analysis was laparoscopic surgery and bowel segment resection (57.7%).

The mean length of hospital stay was 24.6 ± 30.7 days, and the mean length of stay at an intensive care unit was 8.7 ± 18.7 days. In categories 1a and 6, the therapy and diagnosis took place without delay; in category 1b, the mean time to diagnosis was 5.4 ± 4.7 days; and it was challenging to determine the time to diagnosis of an SBP for all other categories because of data leak. Patients collected in category 1b had a longer hospital stay (36.4 ± 39.1 days versus 20 ± 25.4 days for all other categories; *p* < 0.001) and an extended ICU stay (14.5 ± 27.0 days versus 6.4 ± 13.5, respectively; *p* < 0.001). The Clavien–Dindo classification for the 30-day morbidity rate is shown in Table 3.

In sum, 38 patients (14.2%; mean age of 73.3 years) died during hospitalization due to an SBP. The Kaplan–Meier curve of time of survival showed that most of the patients died within one week after diagnosis (Figure 2). The mortality rate of category 1b was not significantly higher than that of category 1a. Patients in category 2 (ischemia) showed significantly higher mortality compared with patients in the other categories (*p* = 0.004).

SBS was established in 16 patients (5.99%). Nearly 30% of patients with SBS suffered from ischemic SBP, and nearly 30% of patients developed the SBS from iatrogenic perforations. In patients with SBS, the in-house mortality rate was 18.8%.

## 4. Discussion

Here, we present a monocentric, retrospective analysis of the etiologies, clinical presentation, therapeutic modalities, and outcomes for SBP, which is a rare event that has relevant impacts on morbidity and mortality. In our analysis, the overall 30-day mortality rate was 14.2%. Especially in patients with ischemic cause (category 2), the SBP mortality rate was significantly increased. Musch et al. reported on an overall mortality rate of 33% for SBP [18]. Klar et al. reported that 10% of patients over 70 years of age with an acute abdomen have mesenteric ischemia [19]. Thus, this is a not-uncommon and clinically severe disease that is often fatal even without the additional presence of SBP and explains the high mortality in category 2 of the presented study results. An analysis of morbidity was performed using the Clavien–Dindo classification [20]. Most of previous reviews or cohort analyses did not use this or any classification. For the further analysis of this topic, a common reporting system would be desirable.

The rate of SBS in the analyzed patient’s cohort was nearly 6%. Any small bowel resection is associated with a risk of developing an SBS. In our analysis, the predominant therapy for SBP was segmental bowel resection, which also led to an increased risk for SBS [3].

Our results showed that iatrogenic perforations (41.9%) were the main causes for small bowel perforations. In other studies, ischemic-obstructive causes or inflammatory-associated perforations were found to be the main origins of small bowel perforations [2,7,21]. We explain the different rates with geographic facts. In some countries, infectious diseases such as typhus or Ruhr (Shigella dysentery) are more common than in Germany. On the other hand, endoscopic, CT, and ultrasound-guided interventions are more common causes for iatrogenic perforations in countries with highly developed medical care standards. However, there are many possible causes for small bowel perforations. More important for outcomes is the grade of the underlying disease (especially for categories 2 and 3) and the time of detection. Ahmed et al. postulated that a delay of up to 4 h from perforation to treatment may have no impact on morbidity or mortality. In a systematic review of bowel injuries in gynecologic laparoscopy, Llarena et al. found that primarily detected and treated SBPs do not worsen the situation of patients [22]. On the other hand, Faria et al. reported a mortality rate of 50% in patients with a delayed diagnosis and therapy of more than 24 h [11].

We present a classification system depending on the etiology and time of perforation for SBP. This classification can be used for further investigations of SBP and for daily clinical work in interdisciplinary settings.

There are currently no common therapeutic recommendations for SBP, though different treatment options exist depending on the localization of SBPs. For duodenal perforations, such as ERCP-related perforations, primary endoscopic treatment could be performed [10,23]. Clipping, the placement of stents, or endoscopic negative pressure therapy are endoscopic treatment modalities for duodenal perforations. Endoscopic negative pressure therapy has especially been shown to result in good clinical courses and could be used to avoid surgery in selected cases [15,16]. Jejunal and ileal perforations are not sufficiently reached during endoscopy.

A clinical scoring system used to stratify patients to possible therapeutic options could be advantageous. In the algorithm presented in Figure 3 for diagnostic and therapeutic steps, a patient’s clinical presentation is measured with the Pittsburgh Perforation Severity Score (PPSS) [24]. This score was created to stratify patients with esophageal perforations for surgical or non-surgical treatment concepts. We believe that this score is transferable to patients with SBP. We present this algorithm for clinical use in cases of suspected SBP.

We are aware of the limitations of this retrospective, monocentric, and non-randomized case series. The single-center design also represents a source of bias. The effectiveness of the presented algorithm and new treatment modalities such as the endoscopic options should be analyzed in prospective studies. However, to the best of our knowledge, this analysis included the highest number of patients with SBP in literature so far.

## 5. Conclusions

To conclude, we have presented a comprehensive analysis of more than 250 patients treated for SBP over a ten-year period, suggested a classification system of SBP according to the etiology and time of diagnosis (Table 1), and described an algorithm of diagnostic and therapeutic steps for treatment. New treatment options are included in this algorithm. The usability of the classification system and the efficacy of the algorithm need to be evaluated in prospective studies.

## Figures and Tables

**Figure 1 jcm-11-05748-f001:**
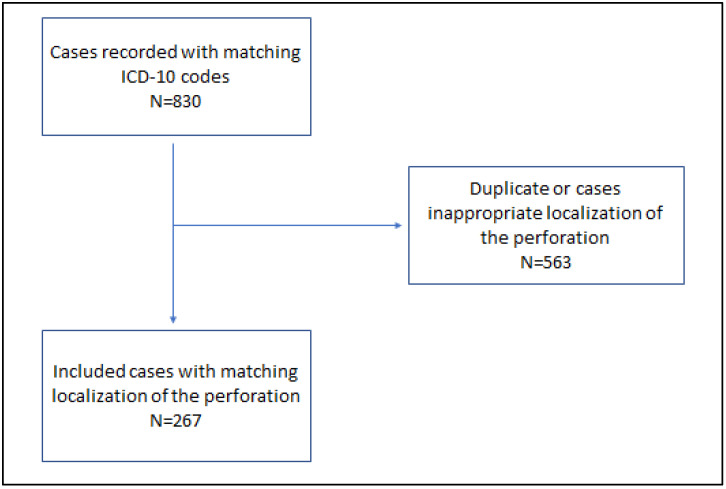
Flowchart of the searching process.

**Figure 2 jcm-11-05748-f002:**
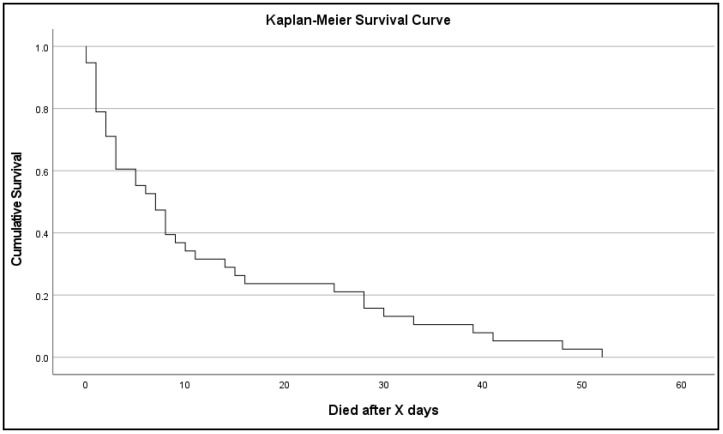
Survival curve of patients with SBP and associated mortality (*n* = 33).

**Figure 3 jcm-11-05748-f003:**
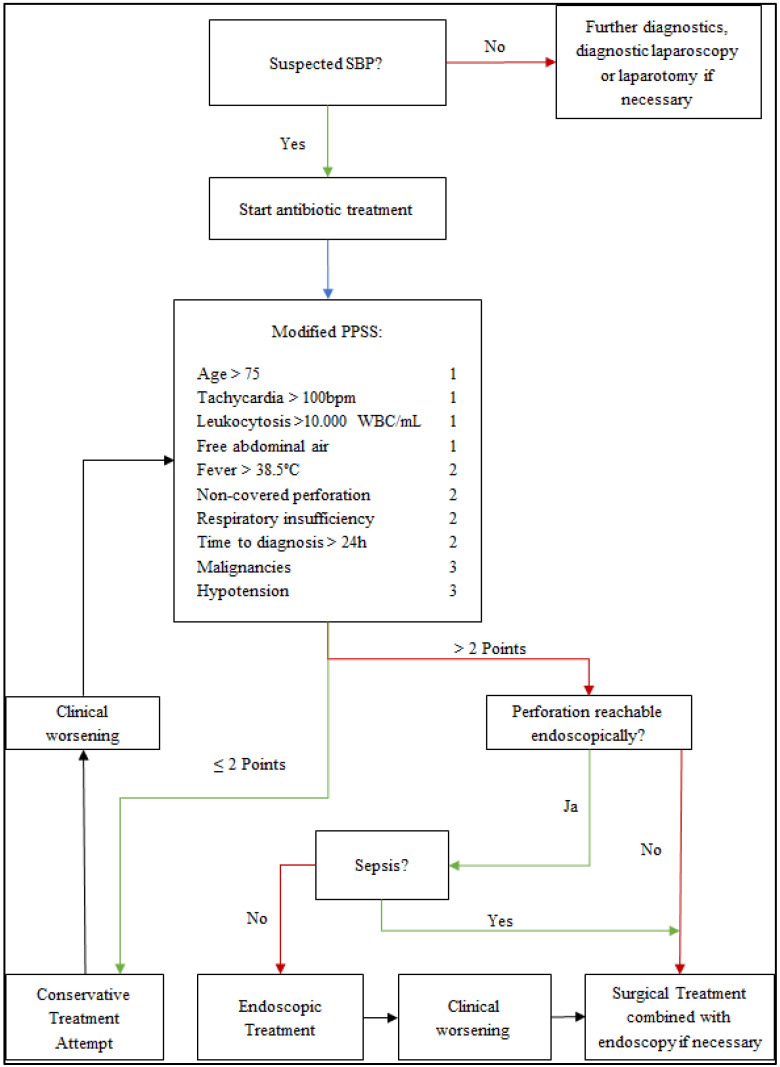
Algorithm for diagnosis and treatment for patients with suspected small bowel perforations. SBP = small bowel perforation; PPSS = Pittsburgh Perforation Severity Score [24].

**Table 1 jcm-11-05748-t001:** Classification of pathogenesis for small bowel perforations with respective number of cases, mean age of patients, and mortality rate.

Category	Causes	Number of Cases (%)	Mean Age (Years)	Number of Death Patients	Mortality Rate (%)
1	(1a) Directly diagnosed and Treated iatrogenic perforation	37 (13.86)	66 (±15)	2	0.75
	(1b) Time delayed diagnosed and treated iatrogenic perforation	75 (28.09)	61 (±16)	11	4.12
2	Ischemic perforation	55 (20.59)	77 (±25)	15	5.62
3	Malignant perforation	45 (16.85)	62 (±16)	7	2.62
4	Inflammatory perforation	22 (8.24)	57 (±12)	1	4.54
5	Diverticula-associated perforation	12 (4.49)	78 (±18)	0	-
6	Traumatic perforations	12 (4.49)	33 (±15)	1	0.37
7	Foreign-body-associated perforation	5 (1.87)	65 (±19)	0	-
8	Unknown cause of perforation	4 (1.49)	70 (±0.4)	1	0.37

**Table 2 jcm-11-05748-t002:** List of therapeutic procedures for SBP.

Mode of Therapy	Number of Cases	%
Laparotomy and perforation over-sewing	72	27
Laparotomy and segment resection	154	57.7
Primary laparoscopy and change for laparotomy with over-sewing	12	4.5
Primary laparoscopy and change for laparotomy with segment resection	7	2.6
Laparoscopic over-sewing	2	0.7
Laparoscopic segment resection	5	1.9
Drainage tube placement	3	1.1
Endoscopic clipping	3	1.1
Conservative treatment	9	3.4

**Table 3 jcm-11-05748-t003:** Analysis of morbidity and mortality rate of small bowel perforations according to the Clavien–Dindo classification.

Clavien–Dindo Classification	Category of Small Bowel Perforations Genesis (According to Table 2)								
	1a	1b	2	3	4	5	6	7	8
CDC 1 (number of cases)	19	17	11	15	11	8	3	5	1
CDC 2 (number of cases)	4	6	4	10	2	1	0	0	0
CDC 3 (number of cases)	6	9	6	6	1	1	1	0	1
CDC 4 (number of cases)	6	34	21	8	7	2	7	0	1
CDC 5 (number of cases)	2	9	13	6	1	0	1	0	1

CDC = Clavien–Dindo classification of 30-day morbidity; CDC 1 = normal course; CDC 2 = need of medication, parenteral feeding, or transfusion therapy; CDC 3 = need of further interventions; CDC 4 = need of therapy on ICU with single or multiorgan failure; CDC 5 = dead 30 days after diagnosis. Categories of small bowel perforations: 1a = directly diagnosed and treated iatrogenic perforation; 1b = iatrogenic perforation with delayed diagnosis and treatment; 2 = ischemic perforation; 3 = malignant perforation; 4 = inflammatory perforation; 5 = diverticula-associated perforation; 6 = traumatic perforation; 7 = foreign-body-associated perforation; 8 = unknown cause of perforation.

## Data Availability

Data available on request.

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
