# Peer review of "Classification und Treatment Algorithm of Small Bowel Perforations Based on a Ten-Year Retrospective Analysis"

_jcm, 2022, doi:10.3390/jcm11195748_

Round 1
Reviewer 1 Report
I have reviewed in detail the paper entitled: “Small Bowel Perforations: Etiology, Treatment and Outcome”. This is an interesting study on an important subject related to the possible contribution of a time of diagnosis and grade of underlying disease in SBP and perforation-associated complications. The introduction has been well written, they provide a detailed explanation of SBP and its main characteristics. Nevertheless, the scope of their study has not clearly specified in the abstract and introduction.
In line 52 they did not description of CT (it is essential for an audience unfamiliar with the topic)
They need to improve the resolution of Figures 1 and 3.
They need to improve the results section and give a better description for the readers; it is complicated to follow the narrative and the relation with figures and table, for example in lines 117 to 122 “Length of hospital stay was in mean 24.6 days with a mean stay at the intensive care unit of 6.4 days. It was challenging to determine the time to diagnosis of a SBP except the category 1b. In the 75 patients collected in category 1b the mean time to diagnosis was 4.0 ± 4.7 days. In patients collected in category 1b mean hospital stay was 36.4 days (p<0.001) with 14.5 days in mean at the intensive care unit (p<0.001). Clavien-Dindo-Classification for 30-days morbidity rate is shown in Table 3.” There is no figure or table that includes the data that they described with respect to the length of hospital and/or intensive care unit stay. It is not clear which groups have been compared. On the other hand, in the wording of the results in table 3, it would be convenient to clarify the importance of this classification.
In lines 135 to 139: “In sum 38 patients (14.2%, mean age 73.3 years) died while hospitalization associated with a SBP. The most of them died until the first week after the diagnosis of a small bowel perforation. See the Kaplan-Meier-Curve of time of survival in Figure 2. Mortality rate of category 1b was not significantly higher than in category 1a, but mortality rate of patients in category 2 (ischemia) was significantly higher (p=0,008).” They compared the rate of survival of the different groups, however, they did not show them in figure 2.
In the discussion, they should include a paragraph about the potential benefits of the proposed algorithms.
Author Response
I have reviewed in detail the paper entitled: “Small Bowel Perforations: Etiology, Treatment and Outcome”. This is an interesting study on an important subject related to the possible contribution of a time of diagnosis and grade of underlying disease in SBP and perforation-associated complications. The introduction has been well written, they provide a detailed explanation of SBP and its main characteristics. Nevertheless, the scope of their study has not clearly specified in the abstract and introduction.
In line 52 they did not description of CT (it is essential for an audience unfamiliar with the topic)
Thanks for this remark, we highlighted the used abbreviation of CT in this position.
They need to improve the resolution of Figures 1 and 3.
Thanks for this remark, the figures are improved now.
They need to improve the results section and give a better description for the readers.
It is complicated to follow the narrative and the relation with figures and table, for example in lines 117 to 122 “Length of hospital stay was in mean 24.6 days with a mean stay at the intensive care unit of 6.4 days. It was challenging to determine the time to diagnosis of a SBP except the category 1b. In the 75 patients collected in category 1b the mean time to diagnosis was 4.0 ± 4.7 days. In patients collected in category 1b mean hospital stay was 36.4 days (p<0.001) with 14.5 days in mean at the intensive care unit (p<0.001). Clavien-Dindo-Classification for 30-days morbidity rate is shown in Table 3.” There is no figure or table that includes the data that they described with respect to the length of hospital and/or intensive care unit stay. It is not clear which groups have been compared. On the other hand, in the wording of the results in table 3, it would be convenient to clarify the importance of this classification.
We improved the result section according this remark, tables and text are updated, so data and compared groups are clarified now.
In lines 135 to 139: “In sum 38 patients (14.2%, mean age 73.3 years) died while hospitalization associated with a SBP. The most of them died until the first week after the diagnosis of a small bowel perforation. See the Kaplan-Meier-Curve of time of survival in Figure 2. Mortality rate of category 1b was not significantly higher than in category 1a, but mortality rate of patients in category 2 (ischemia) was significantly higher (p=0,008).” They compared the rate of survival of the different groups, however, they did not show them in figure 2.
Thanks for this remark. We highlighted the mortality rate of category 2 in the result section in contrast to the overall mortality rate shown in Figure 2.
In the discussion, they should include a paragraph about the potential benefits of the proposed algorithms.
Thanks for this remark, we included the following paragraph: “We present a first classification depending on etiology and time of perforation for SBP. This classification can be used for further investigations on the topic of SBP and for daily clinical work in an interdisciplinary setting.”
Reviewer 2 Report
The manuscript present a comprehensive analysis of patients treated with SBP in a ten-202 year period. But the innovation point and the purpose is not clear. Also, Some other detailed issues are as follows:
1. This study is a retrospective, monocentric, code-related data analysis of patients with a small sample patients of the small bowel perforations. The title is too general and unclear, needs to be revised.
2. In the first paragraph of the introduction, the structure and function of the small intestine are well known, so there is no need to focus on it. It should first introduce the epidemiology and etiology of small intestinal perforation. In addition, the purpose of this study is not clear. Introduction must be rewritten.
3. Line 29:m2
4. Figure1 and 3 are not clear enough, need to been adjusted with high clearance and resolution.
5. “mean age 62.6” Average Age? Values should expressed as Mean ±SD, also Standard deviation of age is missing in Table 1.
Author Response
The manuscript present a comprehensive analysis of patients treated with SBP in a ten-202 year period. But the innovation point and the purpose is not clear. Also, Some other detailed issues are as follows:
Thanks for this remark, we included the following paragraph: “We present a first classification depending on etiology and time of perforation for SBP. This classification can be used for further investigations on the topic of SBP and for daily clinical work in an interdisciplinary setting.”
- This study is a retrospective, monocentric, code-related data analysis of patients with a small sample patients of the small bowel perforations. The title is too general and unclear, needs to be revised.
Thanks for this remark, we changed the title into: “Classification und treatment algorithm of Small Bowel Perforations based on a ten-year retrospective analysis”
- In the first paragraph of the introduction, the structure and function of the small intestine are well known, so there is no need to focus on it. It should first introduce the epidemiology and etiology of small intestinal perforation. In addition, the purpose of this study is not clear. Introduction must be rewritten.
Thanks for this point. We shorted the introduction and focused now on the clinical relevant details, as well as, we pointed out the aim of this study.
- Line 29:m2
We deleted this part of the introduction and shorted it with focus on clinical aspects. We figured out the aim of this study.
- Figure1 and 3 are not clear enough, need to been adjusted with high clearance and resolution.
Thanks for this remark, we submitted the figures as tif. for the publication process.
- 5. “mean age 62.6” Average Age? Values should expressed as Mean ±SD, also Standard deviation of age is missing in Table 1.
We revised Table 1 and added the range in the main text.
Round 2
Reviewer 2 Report
Accept in present form.